# How to Reach Green Word of Mouth through Green Trust, Green Perceived Value and Green Satisfaction

**Jose Antonio Román-Augusto, Camila Garrido-Lecca-Vera, Manuel Luis Lodeiros-Zubiria** \* and **Martin Mauricio-Andia**

Communications Faculty, Universidad Peruana de Ciencias Aplicadas (UPC), Lima 15023, Peru
\* Correspondence: pccmmlod@upc.edu.pe

**Abstract:** The production and consumption of green food products have become hot topics in marketing. Companies are implementing marketing strategies such as green perceived value, green trust, and green satisfaction to guarantee green word of mouth. An online questionnaire distributed through social media was used to collect the data. The sample consists of 297 people. The 297 responses were coded and analysed with the Software Smart-PLS. The data described include the sample sociodemographic profile, the descriptive analysis of all items, the reliability and validity of the measures of the reflective model and the evaluation of the results of the structural model. Four hypotheses included in the PLS-SEM proposed were validated for a *p*-value of 0.001. The results confirmed the influence of green perceived value on green trust and green satisfaction. Moreover, the results highlight that green satisfaction and green trust influence green word of mouth.

**Keywords:** green products; green perceived value; green trust; green satisfaction; green word of mouth





## 1. Summary

The use and consumption of green products have boomed in the last years because of the increasing concern about environmental protection [1] and new lifestyles, where consumers prioritise their well-being and health [2]. Companies have responded to this profitable new market segment by investing in developing and selling a wide range of products classified as green [3] and implementing green marketing strategies to promote them.

Some of the most common green marketing strategies implemented search to increase green word of mouth [4] using other variables such as green perceived value, green trust, and green satisfaction [5–7]. Consequently, the data described in this article provided valuable information about how green product consumers evaluated green perceived value, green trust, green satisfaction, and green word of mouth. Regarding green perceived value, the construct is defined as the overall evaluation that customers develop towards a product based on what is delivered and what is received [8–10]. Green trust is conceptualised as the buyer's willingness to trust a green product based on beliefs or expectations about the environmental and health performance capacity of the green product [5,7]. Finally, green satisfaction has been described as the level of pleasurable fulfilment related to consumption in order to satisfy the environmental and health desires, or expectations of a customer [7–12], green word of mouth is defined as the extent to which a customer influences friends, family or associates by spreading positive environmental messages about a green product [13].

Five hypotheses were formulated based on the previous literature. The first hypothesis postulated that green perceived value positively influences green trust, as Karatu and Mat [14], Lutfie and Marcelino [15] and Pahlevi and Suhartanto [16] found in their research. The second hypothesis posited that green perceived value positively influences green word of mouth. This hypothesis was supported by the indirect effect of green perceived value on green word of mouth found by Roman-Augusto et al. [17]. The third hypothesis was based

on the findings of Chen [18], Pahlevi and Suhartanto [16] and Wang (2022) [19]. The hypothesis postulates that green perceived value positively influences green satisfaction. Green trust positively influences green word of mouth was the fourth hypothesis. The hypothesis was based on the previous findings of customers by Issock et al. [20] Hameed et al. [21] and Suhartanto et al. [22]. The final hypothesis was established following the affirmations of Issock et al. [20]. The hypothesis proposed that green satisfaction positively influences green word of mouth.

Moreover, the tables and figures included in this article describe the sociodemographic profile of the sample (see Table 1) and the relationship between the four variables. Furthermore, the results confirm that green perceived value positively influences the green satisfaction and green trust of the shoppers of green food products. Additionally, the results show that green satisfaction and green trust influence green word of mouth.

**Table 1.** Demographic profile of participants (*n* = 297).

| Demographic | *n* | % | Demographic | *n* | % |
|---|---|---|---|---|---|
| **Age** | | | **Gender** | | |
| Under 18 | 9 | 3% | Female | 189 | 64% |
| 18 to 25 years old | 147 | 49% | Male | 108 | 36% |
| 26 to 35 years old | 95 | 32% | Mode | Female | |
| 36 to 45 years old | 40 | 13% | **Frequency of purchase** | | |
| 46 and over | 6 | 2% | Weekly | 52 | 18% |
| Mode | 18 to 25 years old. | | More than once a month | 59 | 20% |
| **Place of purchase** | | | Monthly | 134 | 45% |
| Eco-fair | 58 | 20% | Infrequently | 52 | 18% |
| Specialised shops | 95 | 32% | Mode | Monthly | |
| Online websites | 54 | 18% | **Products** | | |
| Supermarkets | 90 | 30% | Dairy | 102 | 15% |
| Mode | Specialised shops | | Snacks (cereal bars, biscuits) | 175 | 26% |
| **Occupation** | | | Health drinks | 84 | 12% |
| Student | 102 | 34% | Oils and vinegars | 96 | 14% |
| Employee | 126 | 42% | Granola and corn flakes | 154 | 23% |
| Self-employed | 59 | 20% | Other | 62 | 9% |
| Unemployed or retired | 10 | 3% | Mode | Snacks (cereal bars, biscuits) | |
| Mode | Employee | | | | |

The sample was of 297 people who answered an online questionnaire. The constructs were measured by adapting scales previously used in the green literature and analysed using a partial least square structural equation model (PLS-SEM). The model proposed included five hypotheses, which were checked using the software Smart-PLS [23]. A Bootstrapping of 10.000 confirmed that the five hypotheses were accepted for a *p*-value of 0.005, while only four were supported for a *p*-value of 0.01 or a more restrictive *p*-value of 0.001.

## 2. Data Description

To collect the data, an online questionnaire was designed. Table 1 shows the demographic profile of the participants. From the total sample, 189 were female, and 108 were male. In terms of age range, the mean was from 18 to 25 years old (49%), followed by 26 to 35 years old (32%). A majority of the respondents affirmed that they were employees, with 126 answers, and in second place were students, with 102 responses. Regarding the purchase behaviour of the sample, supermarkets (30%), specialised shops (32%) and ecofairs (20%) were the favourite retailers to acquire green food products. Snacks was the food category that got more answers (26%) when the sample was asked what green food categories they purchase most frequently, second was granola and cornflakes. Finally, the sample stated that in most cases they bought green food products monthly (45%) or more than once per month (20%).Table 2 shows the descriptive results mean (M) and standard deviation (SD) of all items evaluated. Moreover, Table 2 includes the data needed to evaluate the reliability and validity of the measures of the reflective model proposed. According

to Hair et al. and Sarstedt et al. [24–26], the statistics of loadings, Alpha Cronbach's ($\alpha$), Convergence Validity (CR), Rho A ($\rho$A) must all be above 0.7. The results were optimal since all results were above the minimum requested. Alpha Cronbach's range was between 0.72 (GPV) and 0.84 (GT). Convergence Validity results (CR) were 0.84 (GPV), 0.89 (GT), 0.85 (GS) and 0.88 (GW). Finally, the average variance started (AVE) indicators fitted well with the request suggested by Hair et al. [27]. The lowest AVE was 0.64 (GPV), which is well above the minimum recommended of 0.5.

**Table 2.** Items Descriptive Analysis—Reliability and validity of the measures Reflective Model—Loadings.

| Construct | Indicator | M | SD | Loadings | $\alpha$ | $\rho$ A | CR | AVE |
|---|---|---|---|---|---|---|---|---|
| Green Perceived Value (GPV) Pahlevi. M.R.. Suhartanto. D. (2020) [15] | GPV1 | 4.16 | 0.72 | 0.70 | 0.72 | 0.74 | 0.84 | 0.64 |
| | GPV2 | 4.39 | 0.67 | 0.83 | | | | |
| | GPV3 | 4.43 | 0.66 | 0.86 | | | | |
| Green Trust (GT) Pahlevi. M.R.. Suhartanto. D. (2020) [15] | GT1 | 4.26 | 0.75 | 0.79 | | | | |
| | GT2 | 4.15 | 0.72 | 0.81 | 0.84 | 0.84 | 0.89 | 0.68 |
| | GT3 | 3.91 | 0.90 | 0.84 | | | | |
| | GT4 | 4.05 | 0.84 | 0.86 | | | | |
| Green Satisfaction (GS) Pahlevi. M.R.. Suhartanto. D. (2020) [15] | GS1 | 4.28 | 0.62 | 0.79 | 0.73 | 0.73 | 0.85 | 0.65 |
| | GS2 | 4.25 | 0.61 | 0.85 | | | | |
| | GS3 | 4.12 | 0.80 | 0.76 | | | | |
| Green Word of Mouth (GW) Ahmad. W.. Zhang. Q. (2020) [16] | GW1 | 4.12 | 0.76 | 0.71 | 0.82 | 0.83 | 0.88 | 0.65 |
| | GW2 | 4.22 | 0.62 | 0.81 | | | | |
| | GW3 | 4.25 | 0.69 | 0.86 | | | | |
| | GW4 | 4.21 | 0.68 | 0.82 | | | | |

The final stage to evaluate the reliability and validity of the measures of the reflective model was to discard any problems with the discriminant validity of the model. Tables 2 and 3 show the results of the two most common statistical analyses to evaluate the discriminant validity. Table 2 includes the results of the Fornell–Larcker Criterion and Table 3 shows the results of the Heterotrait–Monotrait Ratio (HTMT). Both tables indicate that the reflective model had a correct discriminant validity. since in the results of the Fornell–Larcker Criterion (see Table 3). the first figure of each column was higher than the following figures in the same column. and in the HTMT (see Table 4). because all ratios were under the conservative figure of 0.85 [26].

**Table 3.** Results of Fornell–Larcker Criterion–Discriminant Validity.

| | GPV | GS | GT | GW |
|---|---|---|---|---|
| GPV | 0.80 | | | |
| GS | 0.53 | 0.80 | | |
| GT | 0.63 | 0.66 | 0.82 | |
| GW | 0.51 | 0.63 | 0.60 | 0.81 |

**Table 4.** Heterotrait–Monotrait Ratio (HTMT)—Discriminant Validity.

| | GPV | GS | GT | GW |
|---|---|---|---|---|
| GPV | | | | |
| GS | 0.73 | | | |
| GT | 0.80 | 0.84 | | |
| GW | 0.66 | 0.81 | 0.71 | |

The final stage to evaluate the data in a PLS-SEM was to evaluate the results of the structural model. The results are presented in Figure 1 and Table 5. First. the $R^2$ was calculated. The variance extracted—$R^2$ was 0.40 (GT). GS (0.28) and 0.47 (GW). which can be considered low in all three cases. The five hypotheses were tested using a Bootstrapping

of 10.000 samples and a *p*-value of 0.05. Table 5 includes the results of the path coefficient calculations. The highest path coefficient was between green trust and green perceived value ($\beta = 0.63$). and the relationship between green satisfaction and green perceived value was the second-highest path coefficient ($\beta = 0.53$). Moreover. the path coefficients between green trust. green perceived value. and green satisfaction. with green word of mouth. were $\beta = 0.24$. $\beta = 0.15$ and $\beta = 0.39$. respectively.

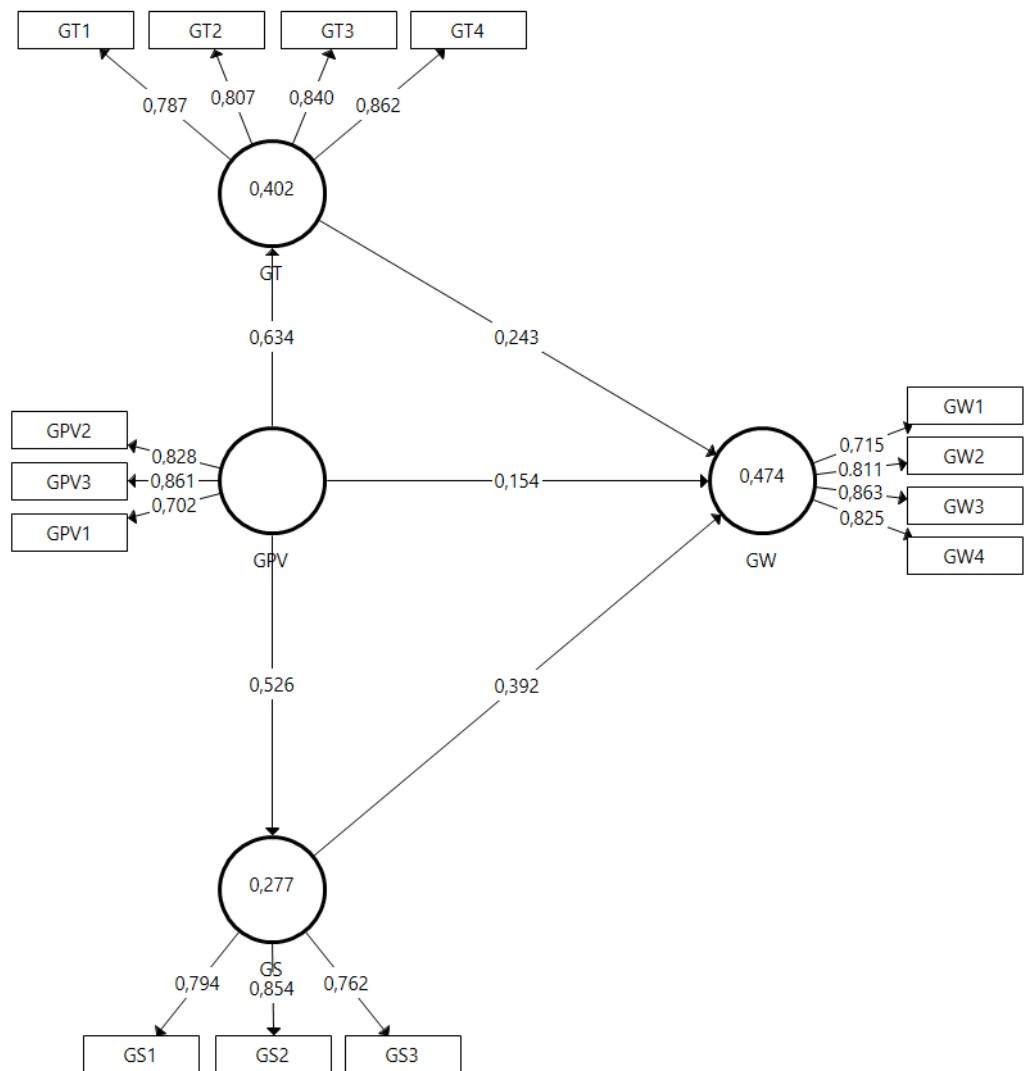

**Figure 1.** Results of the structural model—$R^2$—Outer Loadings and Path Coefficients.

**Table 5.** Results Estimation of the Structural Model—*p*-Values—Path Coefficients.

| Criterion | Predictor | β | t-Value | $R^2$ | Inner VIF Values |
|---|---|---|---|---|---|
| Green Word of Mouth (GW) | Green Perceived Value (GPV) | 0.15 | 2.29 | 0.47 | 1.73 |
| | Green Trust (GT) | 0.24 | 3.44 | | 2.24 |
| | Green Satisfaction (GS) | 0.39 | 5.15 | | 1.85 |
| Green Satisfaction (GS) | Green Perceived Value (GPV) | 0.53 | 9.15 | 0.28 | 1.00 |
| Green Trust (GT) | Green Perceived Value (GPV) | 0.63 | 13.12 | 0.40 | 1.00 |

Table 6 collects the information gathered to confirm or reject the hypotheses proposed. A bootstrapping with 10.000 samples and * $p < 0.05$; ** $p < 0.01$; *** $p < 0.001$ (two-tailed) was executed. The results supported H1. H2. H3 and H5. while H4 was rejected. All four hypotheses were accepted for ** $p < 0.01$ and *** $p < 0.001$ (two-tailed). If hypothesis H4 had been evaluated with a $p < 0.05$ (two-tailed). it would have been accepted. The signs of

all relationships were positive. The results allow a conclusion that. excluding the green perceived value. the other two variables. green trust and green satisfaction. influence the creation of green word of mouth. Nevertheless. green perceived value positively influences green trust and green satisfaction.

**Table 6.** Results Estimation of the Structural Model—*p*-Values—Path Coefficients.

|   | Hypotheses | *p*-Value | Sign | Result |
|---|---|---|---|---|
| H1 | Green Perceived Value (GPV) → Green Trust (GT) | 0.00 | + | Supported |
| H2 | Green Perceived Value (GPV) → Green Satisfaction (GS) | 0.00 | + | Supported |
| H3 | Green Trust (GT) → Green Word of Mouth (GW) | 0.00 | + | Supported |
| H4 | Green Perceived Value (GPV) → Green Word of Mouth (GW) | 0.02 | + | Not Suported |
| H5 | Green Satisfaction (GS) → Green Word of Mouth (GW) | 0.00 | + | Supported |

## 3. Methods

The data needed were collected using an online questionnaire. since this was considered the most convenient method to reach the population of the study. The questionnaire was distributed through social media platforms such as Facebook. WhatsApp. and Instagram. The authors used a non-probabilistic method. snowball. to reach the study's population. since there was not a census of green product consumers in Peru. the country where the research took place. Nevertheless. the research data collection was made following a rigorous protocol to guarantee that only people within the study's population responded to the online questionnaire. QuestionPro [28] was the online service provider used to distribute the survey and store the answers.

The study population considered consumers of green food products in Peru. The final sample consisted of 297 people. Although 404 people participated in the survey. only 297 responders completed the full questionnaire. answering correctly the six demographic and the eleven variable questions included in the research instrument. The instrument was divided into two sections. In the first section. six questions were used to define the demographic sample and purchase behaviour of the sample. In the second section. the four constructs were evaluated using Likert scales from 1 to 5. where 1 was equal to completely disagree. while 5 was equal to completely agree. The four constructs evaluated were green perceived value. green trust. green satisfaction. and green word of mouth.

The second part of the instrument was designed to collect data about the five constructs proposed in the research model (see Figure 1). The construct green perceived value was coded as GPV and evaluated with three items based on the scale proposed by Pahlevi and Suhartanto [16]. The second variable was green trust. coded as GT. Four items were considered to measure it. following the criteria of the Green Trust (GT)—Pahlevi and Suhartanto scale [16]. GS was the code used for green satisfaction. Its three items were adapted from the scale developed by Pahlevi and Suhartanto [17]. Finally. green word of mouth was evaluated with four items. based on the scale proposed by Ahmad and Zhang [6]. the variable coded as GW.

It is worth mentioning that the eleven Likert scales were originally written in English. but during the instrument. design processes were translated into Spanish. The eleven items proposed in the research were: GPV1—Organic products provide more benefits than the cost of obtaining them. They are worth it. GPV2—Organic products are more environmentally sound and health-conscious than more conventional products. GPV3—Organic products are more beneficial to my health and the environment than more conventional ones. GT1—I believe organic products have a good reputation because they help our health and the environment. GT2—I believe that organic products are reliable. GT3—I believe in brands that sell organic and ecofriendly products. GT4—I believe that organic products live up to their promises to care for our health and the environment. GS1—Using organic products makes me feel good and satisfied. GS2—I think it is a wise decision to buy organic products. GS3—Overall. I am satisfied with organic products. GW1—Due to their ecofriendly and healthy image. organic products are recommended by other people.

GW2—Due to their environmental and health benefits. organic products are positively recommended by other people. GW3—Due to being environmentally friendly and healthy. organic products have a good reputation. GW4—Due to their environmental and health benefits. organic products receive positive feedback from people.

The responses got from the 297 people were coded and transformed into a .csv file.

Since the model proposed included five hypotheses. a Partial Least Square structural equation model (PLS-SEM) was used. The PLS-SEM was chosen since it is the most appropriate statistical approach to solve complex models as the one proposed in this research [29]. The file was uploaded to the Smart-PLS 3.7.7 [23] software. and from the dataset the PLS-SEM model proposed was drawn (Figure 2) and evaluated with a PLS Algorithm and a Bootstrapping of 10,000 samples. * $p < 0.05$; ** $p < 0.01$; *** $p < 0.001$ (two-tailed) [24,30].

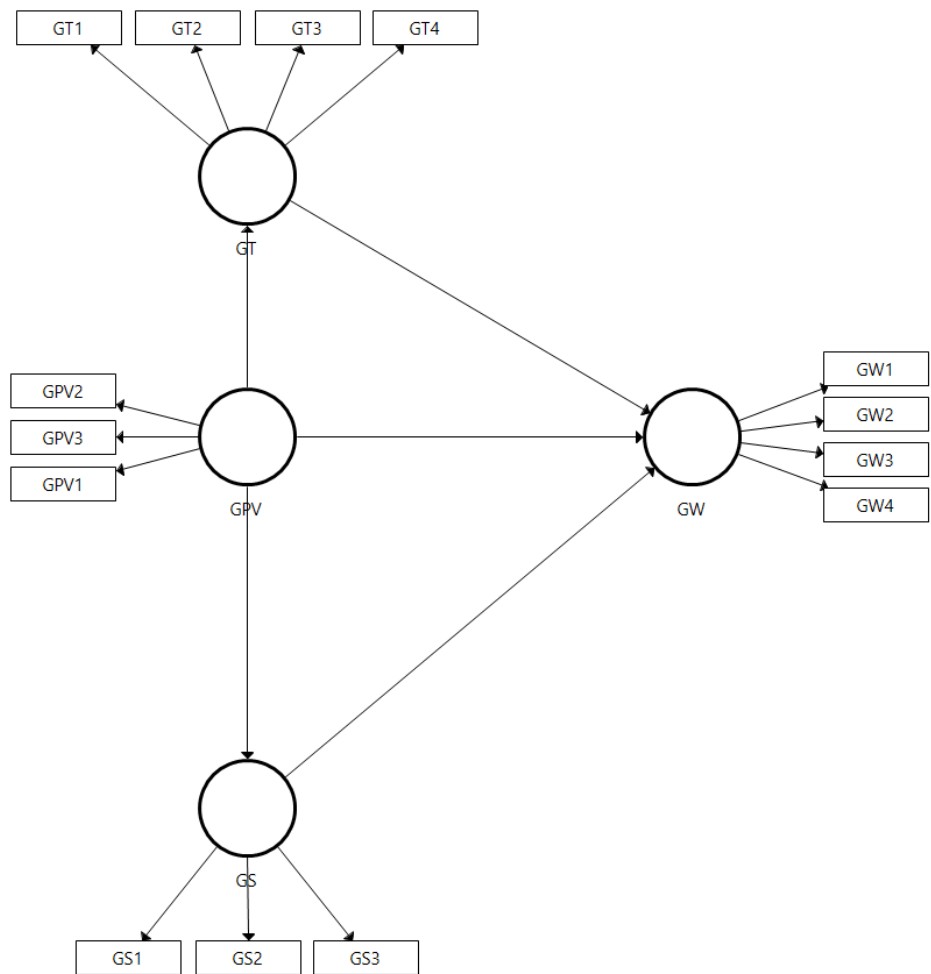

**Figure 2.** PLS-SEM proposed.

## 4. Conclusions

This research shows a series of contributions in the field of Green Products Marketing and contributes to the previous literature developed by Wang et al. [31]. McDougall et al. [9] Issock Issock et al. [20]. Zhang et al. [32] and Román et al. [18]. The first contribution reinforces the importance of green perceived value as a critical variable in developing an effective green marketing strategy. The results have shown that the green perceived value positively influences the creation of green trust and green satisfaction. as previously postulated [9,16]. Moreover. the results confirmed the existence of a direct relationship between green perceived value and green word of mouth and complemented the previous findings of Román et al. [17]. who found an indirect relationship between both constructs

moderated by green trust and green satisfaction. Finally. the results reinforced the previous literature's conclusion and proved the influence of green trust and satisfaction in creating word of mouth [9,33].

The research has some limitations that could be addressed. Firstly. it is worth mentioning that the data were collected using a non-probabilistic method. limiting the universalisation of the results. Secondly. the vast range of green products included in the sample could represent an inconvenience in correctly understanding green customer behaviour since the purchase intention of green products could differ according to the final use and purchase frequency. Consequently. future searches could consider including the purchase intention. the purchase frequency or the product category as dependent variables or as moderators.

**Author Contributions:** Conceptualization. J.A.R.-A., C.G.-L.-V, M.L.L.-Z. and M.M.-A.; methodology. M.L.L.-Z.; software. M.L.L.-Z.; validation. M.L.L.-Z.; formal analysis. M.L.L.-Z.; data curation. M.L.L.-Z.; writing—original draft preparation. M.L.L.-Z.; writing—review and editing. M.L.L.-Z.; supervision. M.L.L.-Z. project administration. M.L.L.-Z.; funding acquisition. M.L.L.-Z. All authors have read and agreed to the published version of the manuscript.

**Funding:** Universidad Peruana de Ciencias Aplicadas/UPC-EXPOST-2022-2.

**Institutional Review Board Statement:** Not applicable.

**Informed Consent Statement:** Informed consent was obtained from all subjects involved in the study. In the online questionnaire. all responders were asked for their permission to collect the data and analyse it globally. No personal or confidential information was asked for or collected during the research process.

**Data Availability Statement:** The dataset of this study can be found at DOI: 10.17632/4m6t8c63k8.1 under a CC BY 4.0 license.

**Acknowledgments:** We would like to thank the Research Directorate of the Universidad Peruana de Ciencias Aplicadas for the support provided to carry out this research work through the UPC-EXPOST-2022-2.

**Conflicts of Interest:** The authors declare no conflict of interest.

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
