# Peer review of "How to Reach Green Word of Mouth through Green Trust, Green Perceived Value and Green Satisfaction"

_data, 2022_

Round 1

Reviewer 1 Report

Dear authors,

the subject of the paper captured my attention as an interesting one. 

I think a paper with such a subject is appropriate to be published. To be published, I consider some improvements should be considered:

- an introductory section where the green concepts included in the paper are to be discussed, with references to previous research and relevant papers, is necessary. Also, some theory related to the correlation between trust, satisfaction, perceived value - would be useful;

- "previously used in the green literature" - references here

- please clearly formulate the hypothesis 

- "descriptive results (mean and standard) deviation" - ?? 

- more details of the type of research are needed 

- a section of Discussions and Conclusions is absolutely necessary + further research directions (in case)

- also, some limitations of the research are important to be presented

Author Response

Point 1: I think a paper with such a subject is appropriate to be published. To be published, I consider some improvements should be considered:

- an introductory section where the green concepts included in the paper are to be discussed, with references to previous research and relevant papers, is necessary. Also, some theory related to the correlation between trust, satisfaction, perceived value - would be useful. Previously used in the green literature" - references here. -Please clearly formulate the hypothesis

Response 1: The definitions of each variable was included in the paper. The hypotheses were included and supported by the previous literature.

Point 2: - "descriptive results (mean and standard) deviation". More details of the type of research are needed

Response 2: All descriptive indicator abbreviations are now linked to the original indicator. An explnaion of the research type was included. An explanation of the PLS-SEM method can now be found in the article.  

Point 3: - - a section of Discussions and Conclusions is absolutely necessary + further research directions (in case). Also, some limitations of the research are important to be presented.

Response 3: A new section,“Conclusions” has been included. The section highlights the investigation's most important findings and compares them to the previous literature. Moreover, limitations and future directions have been considered. 

Reviewer 2 Report

The work is direct, rapid and easy to read, the data and methods are good however the work can be improved in terms of Introduction (summary), support from the literature and conclusions.

1- add more support from the literature 

2- add conclusions 

Author Response

Point 1: Add more support from the literature

Response 1: The definitions of each variable was included in the paper. The hypotheses were included and supported by the previous literature.

Point 2: Add conclusions

Response 3: A new section,“Conclusions” has been included. The section highlights the investigation's most important findings and compares them to the previous literature. Moreover, limitations and future directions have been considered. 

Round 2
